# Microfluidic Fabrication and Thermal Properties of Microencapsulated N-Hexadecane with a Hybrid Polymer Shell for Thermal Energy Storage

**DOI:** 10.3390/ma15103708

**Published:** 2022-05-22

**Authors:** Luxi Yang, Linchuan Dai, Lu Ye, Rui Yang, Yangcheng Lu

**Affiliations:** Department of Chemical Engineering, Tsinghua University, Beijing 100084, China; luxi_yang@foxmail.com (L.Y.); dlc17@mails.tsinghua.edu.cn (L.D.); jennsie16@163.com (L.Y.)

**Keywords:** phase change materials, microencapsulation, microfluidics, thermal energy storage

## Abstract

In this study, a strategy based on microfluidic method is developed toward a facile fabrication of phase change material microcapsules with uniform and controllable particle size as well as high encapsulation ratio and thermal stability. N-hexadecane, as a phase change material, was successfully encapsulated by a hybrid shell of poly (methyl methacrylate) and polyurea. The fabrication process includes the following three steps: (1) Formation of oil-in-water droplets with uniform micron size in the microfluidic chip; (2) formation of the first polyurea shell to encapsulate droplets by fast interfacial polymerization when the droplets pass through the coiled transport microchannel; and (3) completion of free radical polymerization of methyl methacrylate inside the microspheres by heating to form the hybrid microcapsule shell. The average size, encapsulation ratio, and phase change enthalpy of microcapsules changed by varying the flow rate of the dispersion phase and raw material composition. The highest melting enthalpy of 222.6 J g^−1^ and encapsulation ratio of 94.5% of the microcapsule were obtained when the flow rates of the continuous and dispersion fluids were 600 μL min^−1^ and 24 μL min^−1^, respectively. It is shown that the phase change material microcapsules were stable after 50 heating/cooling cycles.

## 1. Introduction

Thermal energy storage (TES) systems have received much attention in recent years, due to the increasingly serious energy crisis and environmental problems [1,2]. It is reported that, on a global scale, total primary energy supply was 573 EJ and final consumption was 394 EJ per year, with around 31% energy was lost [3]. TES can balance energy supply and demand and thus guarantee the stable and continuous use of energy as well as enhance the utilization efficiency of solar energy [4]. It is generally accepted that latent heat storage systems, associated with the phase transformation of substances, have higher heat storage density than other TES systems [5]. The most attractive term for the working media of latent heat storage systems is phase-change materials (PCMs), because they do not require external energy input, are low cost, and can meet the heat storage requirements over a wide temperature range [6]. Even with these advantages, leakage and erosion issues seriously restrict the direct application of bulk PCMs unless they are enclosed or encapsulated upon melting or evaporation [2,7].

Microencapsulated PCMs (MPCMs) can avoid the leakage of PCMs, reduce the reactivity of PCMs with the external environment and tolerate volume change as phase change occurs [8,9,10]. Desirable MPCMs should exhibit high thermal performance and have tunable thermal properties to meet the various demands of pragmatic applications involving different and complex environments, calling for a versatile fabrication technology capable of precisely and flexibly controlling the size and structure of MPCMs. Thus far, the following three methods are available to fabricate MPCMs: physical methods such as spray drying [11] and solvent evaporation [12], chemical methods such as emulsion polymerization [13,14] and suspension polymerization [15], and physical−chemical methods such as coacervation [16] and the sol−gel method [17]. However, the above preparation methods have difficulty ensuring the uniformity and controllability of the size of microcapsules, which limits the flexibility and efficiency in a wide variety of applications. The obtained MPCMs cannot guarantee mechanical strength to meet the stability demand of multiple heat storage cycles. Meanwhile, the existing preparation for MPCMs is more in small-scale batch ways, which means the consistency between batches and large-scale production is difficult to achieve. Therefore, it is essential to develop a continuous and stable fabrication method for PCM microcapsules with uniform and controllable particle size, high encapsulation rate of PCM, and high thermal stability.

In recent years, droplet microfluidic technology has provided a versatile platform for the fabrication of uniform micron-sized droplets as emulsions, polymer particle templates, and microcapsule templates [18,19,20,21], since it could precisely handle a small fluid volume and allows accurate control over the formation of droplets. As for preparing MPCMs, microfluidic droplets have been demonstrated as ideal templates with narrow size distribution, high tunability of sizes, and relatively low material cost [22], enabling the MPCMs with tunable thermal properties and high control of thermal performance and multifunctionality.

However, the coalescence of micron-sized droplets in emulsion deteriorating the size control seriously is difficult to evade in the scale-up for production corresponding to the droplet collisions aggravating. A common strategy to inhibit this negative effect is to cure the surface of the droplets as soon as possible, which will demand specific formula in droplets and constrain the cost and composition of MPCMs. In this study, a new strategy based on microfluidic method is proposed toward a facile fabrication of MPCMs with uniform and controllable particle size, as well as long-term thermal stability. N-hexadecane, a PCM, was successfully microencapsulated by a hybrid polymer shell of polyurea (PU) and poly(methyl methacrylate) (PMMA). This microencapsulation was jointly achieved by staged polymerization of interfacial and suspension-like polymerization in an oil-in-water microfluidic droplet. We also explored the preparation conditions of microfluidic droplet templates and MPCMs. The resulting MPCMs have a well-controlled core-shell structure and exhibit high thermal performance, including energy storage capacity, thermal stability and reliability.

## 2. Materials and Methods

### 2.1. Experiment Materials

n-Hexadecane (C16) (98%, purity) was purchased from Meryer; methyl methacrylate (MMA) (CP), isophoronediisocyanate (IPDI) (99%), diethylenetriamine (DETA) (99%), and azodiisobutyronitrile (AIBN) (98%) were purchased from Macklin; pentaerythritol tetraacrylate (PETRA) (>80%) was purchased from TCI; and Tween 80 and Span 80 were purchased from Aladdin. All chemicals were used as received without further purification.

### 2.2. Microfluidic Setup and Microcapsule Fabrication Method

Microcapsules with hybrid polymer shells of PU and PMMA were successively fabricated by interfacial polymerization in a microfluidic chip and suspension-like polymerization in a three-neck flask. The synthetic scheme is shown in Figure 1. The microfluidic chip consisted of two inlet ports, one outlet port, and one T-junction microchannel. The continuous phase flowed through a straight main channel, and the dispersion phase entered the main channel through a side channel perpendicular to the flow [23]. The continuous and dispersed fluids were injected into the microfluidic chip by syringe pumps (Harvard). The inner diameters of the tubing for continuous and dispersion fluids were 0.8 and 0.27 mm, respectively. The MPCM preparation process is shown in Figure 2.

The continuous fluid (aqueous phase) was composed of 3 wt.% aqueous solutions of Tween 80. DETA was added to the continuous fluid. The dispersed phase (oil phase) was composed of C16 with 3 wt.% Span 80. MMA (purified by vacuum distillation), IPDI, AIBN, and PETRA were added to the dispersion phase. The flow rate was 600 μL min^−1^ for the continuous fluid and ranged from 24–60 μL min^−1^ for the dispersed phase.

At the T-junction, the oil phase was sheared into droplets. When the droplets flowed through the coiled transport channel at room temperature in 3 min, interfacial polymerization of DETA and IPDI occurred, and a PU shell formed. The preformed microcapsules were then collected into a three-neck flask placed in a 70 °C water bath to complete suspension-like polymerization under stirring. Finally, microcapsules of C16 encapsulated by a PU/PMMA composite shell were fabricated. The product was then filtered, washed three times with ethanol, and dried in an oven at 60 °C for 24 h for further characterization.

### 2.3. Characterization

Fourier transform infrared spectroscopy (FTIR) analysis of MPCMs was performed using an FT-IR instrument (IR Tracer-100, SHIMADZU Ltd., Kyoto, Japan). Microcapsules were mixed with KBr powder and ground together. Sample disks were measured at room temperature with a resolution of 4 cm^−1^ and a wavenumber range of 4000–400 cm^−1^.

The morphology of the MPCMs was observed using a scanning electron microscope (SEM, TM3000, HITACHI Ltd., Tokyo, Japan). Elemental analysis of the microcapsule shell was performed with an attached energy dispersive spectrometer (EDS).

The thermal properties were studied using a differential scanning calorimetry (DSC) instrument (DSC 214, NETZSCH Ltd., Bavaria, Germany) in a nitrogen atmosphere with a flow rate of 60 mL min^−1^. Samples were heated or cooled between −30 °C and 50 °C at a rate of 10 °C min^−1^. The heat curves in enthalpy change region were analyzed. The encapsulation ratio (C), an important parameter to evaluate the thermal performance, can be calculated using the following Equation (1):(1)$$C=\frac{\Delta H_{m,MEPCM}}{\Delta H_{m,PCM}}\times 100\%$$
where $\Delta H_{m,MEPCM}$ and $\Delta H_{m,PCM}$ are the measured melting enthalpies of MPCM and n-hexadecane, respectively.

Thermogravimetric analysis (TGA) was carried out using a TG 209 thermogravimetric analyzer (NETZSCH Ltd., Bavaria, Germany). TGA was performed from 35 to 1000 °C at a heating rate of 10 °C min^−1^ in a nitrogen atmosphere with a flow rate of 20 mL min^−1^ for all samples.

The cyclic stability of the MPCMs was evaluated by DSC and the refrigerator-oven method. In DSC evaluation, an MPCM was heated to 50 °C and cooled to −30 °C for 50 cycles at 10 °C min^−1^. The encapsulation ratio after each cycle was calculated. In the fridge-oven method, MPCMs were put in an oven set at 35 °C for 2 h and then moved to a fridge set at 0 °C for 2 h. This process was repeated 50 times. The encapsulation ratio after each cycle was detected by DSC.

## 3. Results and Discussion

### 3.1. Microfluidic Fabrication of MPCMs

The FT-IR spectra of pure n-hexadecane, PU, PMMA, and a typical MPCM are exhibited in Figure 3. Peaks at 2923 and 2853 cm^−1^ are identified as C-H stretching vibration bands of n-hexadecane and are also observed in the spectrum of MPCM. The typical characteristic peak of C=O at 1732 cm^−1^ is found in both the MPCM and PMMA spectra. In addition, absorption peaks are observed for the N–H stretching vibration and N–H bending vibration at 3359 and 1558 cm^−1^ in both the MPCM and PU spectra. These results indicate that n-hexadecane is successfully encapsulated in MPCMs. Moreover, the characteristic peak of -NCO at approximately 2260 cm^−1^ can be observed from the MPCM spectrum, corresponding to the residual IPDI.

The SEM photo of a typical MPCM is shown in Figure 4, in which elemental analysis of the MPCM shell was determined by EDS. On the whole, the fabricated MPCMs have spherical shapes, with smooth but dented surfaces (Figure 4a). The dents on the surfaces are attributed to the volume change of the phase change materials in the core when the experimental temperature changes. This also indicates that the MPCMs prepared by this method can withstand the volume change caused by the phase change without breaking down. By breaking a microcapsule, the core-shell structure is proven in Figure 4b. The shell thickness is about 0.4 μm. Elemental analysis of the inner and outer surface of the shell (Figure 4c–e) shows the existence of C, N, and O with different ratios. N% is higher at the outer surface, and O% is higher at the inner surface. This result suggests that although PU and PMMA shells were formed in two successive steps, a hybrid shell other than a double shell was formed. In the first step, the PU shell quickly formed to prevent coalescence between droplets. Then, the MMA monomer diffuses into the PU shell and polymerizes to strengthen the shell.

### 3.2. Effects of Preparation Conditions of MPCMs

#### 3.2.1. Microcapsule Composition

In this study, a hybrid shell was prepared during microfluidic fabrication. The PU shell was used to separate droplets and prevent coalescence in a short time, but the PU shell was too thin to be strong. Then, PMMA was used to strengthen the shell. PU acted as a skeleton structure filled with PMMA. Therefore, the ratio of IPDI to MMA in microcapsule preparation, which relates to the ratio of PU to PMMA in the MPCM shell, was considered. In addition, the effect of the core/shell ratio is also studied. Melting and crystallization curves of microcapsules with various IPDI:MMA:C16 ratios by DSC are shown in Figure 5, and the thermal properties are listed in Table 1. The flow rates of the continuous phase and the dispersed phase are set as 600 and 24 μL min^−1^, respectively.

It is obvious that the melting and crystallization enthalpies of the sample with IPDI:MMA:C16 = 5:25:70 are the lowest among the MPCMs. This results from the low ratio of IPDI to MMA. In this case, a low content of PU could not form a complete shell in the first step, leading to C16 leakage and low encapsulation efficiency. The melting and crystallization enthalpies of the other MPCMs are relatively high, reaching 210.7 and 207.3 J g^−1^ when IPDI:MMA:C16 = 8:22:70. It is interesting that for samples with IPDI:MMA:C16 = 8:12:80, IPDI:MMA:C16 = 10:20:70, and IPDI:MMA:C16 = 8:22:70, the encapsulation ratio is nearly 90%, much higher than the C16 contents of 70% and 80%. This is an additional evidence that IPDI and MMA were not completely polymerized. This is reasonable because MMA is partly soluble in water, and some IPDI in the oil droplet was not reacted, as also shown in Figure 3. The endothermic and exothermic peaks are broad, the melting temperature is higher than 19.5 °C, the phase change temperature of n-hexadecane. This is because PU and PMMA have poor thermal conductivity, resulting in the delay of the phase change.

For the thermal stability of MPCMs, Figure 6 shows the TGA thermograms and derivative thermogravimetry (DTG) thermograms of MPCMs with different raw material compositions. The pristine n-hexadecane starts to lose weight at approximately 145 °C and loses its total weight at approximately 190 °C; however, the beginning and complete weight loss temperatures of PU and PMMA reach approximately 295 and 500 °C, respectively. Therefore, the TGA thermograms are divided into the following three stages: C16 evaporated before 200 °C; encapsulated C16 evaporated in the range of 200–295 °C; the PU/PMMA shell decomposed in the range of 295–500 °C. The mass fractions of MPCMs calculated from TGA thermograms are shown in Table 2. Obviously, microencapsulation can delay the evaporation of C16 by protection from the hybrid polymer shell. In particular, the weight loss of the sample with IPDI:MMA:C16 = 10:20:70 mainly occurs from 200 to 295 °C, showing a strong delay in C16 evaporation. These results suggest that MPCMs have good thermal stability and can be used over a wider range of temperatures.

#### 3.2.2. Flow Rate of the Dispersed Phase

The flow rates of the continuous phase and the dispersed phase are essential to the formation of microfluidic droplets for microcapsules. To study this effect, the flow rate of the continuous phase is fixed at 600 μL min^−1^, and the flow rate of the dispersed phase varies between 24 and 60 μL min^−1^. According to the effects of the microcapsule composition mentioned above, the IPDI:MMA:C16 ratio of MPCM preparation is selected as 10:20:70, which shows good thermal stability.

The encapsulation ratio decreases gradually with increasing dispersed phase flow rate, as shown in Table 3. This means that the polymerization conversion rate decreases, which relates to the density of the initial PU shell. At a low flow rate of the dispersed phase, the continuous phase is able to ensure a sufficient supply of monomers in the water phase to form a relatively dense initial PU shell layer. This dense layer hinders the further polycondensation reaction and causes incomplete polycondensation and residual monomers IPDI in the droplets, which is also proven in Figure 3. IPDI may also hinder the subsequent free radical polymerization of MMA. Since the residual IPDI and MMA will be removed by subsequent washing and drying processes, they will not be included in the calculation of the encapsulation rate. In this case, a relatively high encapsulation rate is achieved. At the same time, a dense layer has better resistance to the liquid–liquid interface shrinkage caused by the mutual dissolution of MMA and water at the polymerization temperature, corresponding to a large particle size. In contrast, when the flow rate of the dispersed phase is high, the supply of the monomers in the water phase is relatively insufficient, forming a less dense initial PU shell. This drives a complete polycondensation reaction and a higher conversion of the free radical polymerization and thus a lower encapsulation rate. Accordingly, the less dense shell shrinks with the shrinkage of the liquid–liquid interface due to the mutual dissolution of MMA and water, and the product particles decrease.

The effect of the dispersed phase flow rate is further demonstrated by morphology observation of MPCM samples by SEM, as shown in Figure 7. It is found that the particle size is indeed large at a low disperse phase flow rate. When the flow rate of the dispersed phase increases to more than 36 μL min^−1^, its effect on the particle size is not obvious. Overall, the MPCMs remain stable in sizes ranging from 80 to 300 μm. The size distribution exhibits some inhomogeneity, arising from the collisional coalescence of droplets before the shell is completely cured. Future work is required to improve the stability of the two-phase flow in the channel following the microfluidic chip.

The flow rate of the dispersed phase also influences the thermal properties of MPCMs during the melting and crystallization process, which was studied by DSC. The melting and crystallization curves are shown in Figure 8, and the thermal properties are listed in Table 3. All MPCMs show broad melting and crystallization peaks because of their large microcapsule size. Typically, MPCMs fabricated by conventional interfacial polymerization with PU-based shell, can achieve encapsulation ratio of 85.28% [24]. In comparison, the encapsulation rates of all samples in this section exceeded 86%, and the enthalpy was over 200 J g^−1^. When the flow rate of the dispersed phase continuously decreases, the melt-ing/crystallization enthalpy increases slightly. Especially, the melting/crystallization enthalpy reaches the maximum (>220 J g^−1^) at 24 μL min^−1^, with the highest encapsulation rate of 94.5%.

### 3.3. Cyclic Stability of MPCMs

The MPCM with IPDI:MMA:C16 = 10:20:70 exhibits good thermal properties. Its cyclic stability was evaluated by DSC and the fridge-oven method, and the encapsulation ratio change is shown in Figure 9. The MPCM exhibits excellent cyclic stability. In the DSC and fridge-oven evaluations, the encapsulation ratio remained nearly unchanged, with a fluctuation of less than 5% after 50 cycles.

## 4. Conclusions

A new strategy based on microfluidic method is developed toward a facile fabrication of MPCMs with uniform and controllable size in microns and high thermal performance. N-hexadecane was successfully microencapsulated by a hybrid polymer shell of PU and PMMA during successive interfacial polymerization and suspension-like polymerization. The flow rates of the dispersed phase influence the size during droplet generation as well as the phase change enthalpy and encapsulation ratio. At flow rates of 600 μL min^−1^ for the continuous phase and 24 μL min^−1^ for the dispersion phase, the highest melting enthalpy of 222.6 J g^−1^ and an encapsulation ratio of 94.5% were achieved. With suitable shell material components, the encapsulation ratio can be as high as approximately 90%. Moreover, our MPCMs have high thermal stability and cyclic stability due to protection from the hybrid shell material after 50 cycles. Therefore, the microfluidic method is a powerful tool for the fabrication of large MPCMs with high thermal properties and good cyclic stability. Further improvements should focus on the stability of two−phase flow in the channel following the microfluidic chip as the next step. At the same time, preparation on a larger scale for technological readiness still remains to be further explored.

## Figures and Tables

**Figure 1 materials-15-03708-f001:**
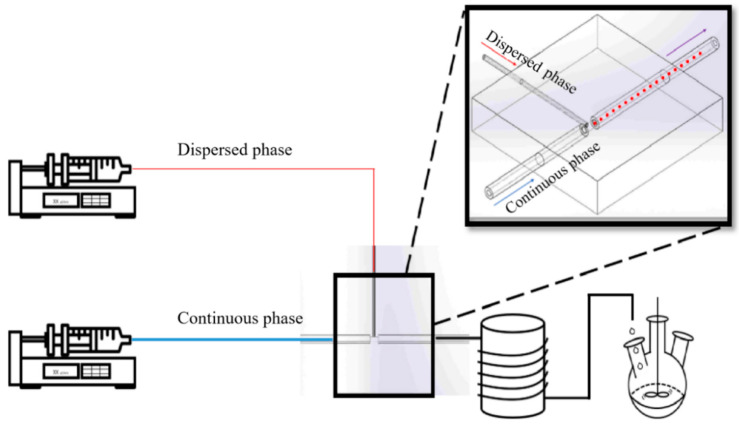
Synthetic scheme for MPCMs by interfacial and suspension-like polymerization.

**Figure 2 materials-15-03708-f002:**
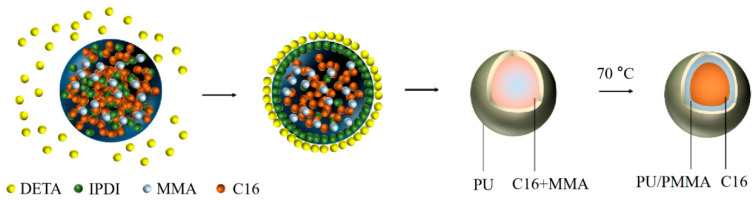
Schematic diagram for MPCM preparation by interfacial and suspension-like polymerization.

**Figure 3 materials-15-03708-f003:**
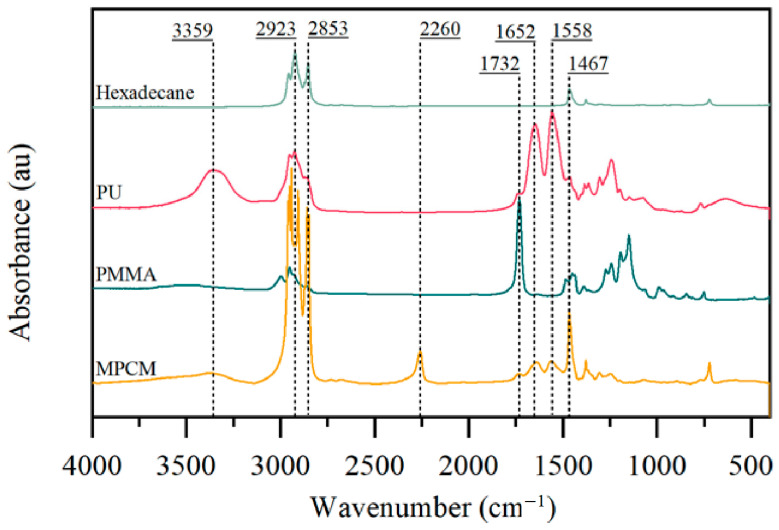
FT-IR spectra of n-hexadecane, PU, PMMA, and MPCM.

**Figure 4 materials-15-03708-f004:**
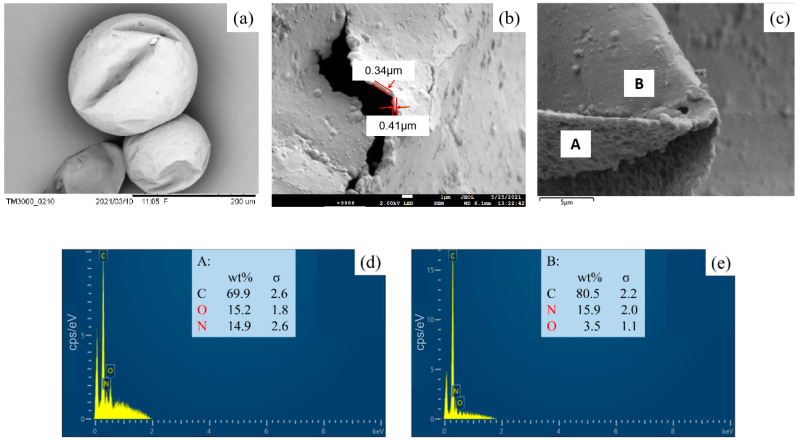
(**a**) Morphology of a typical MPCM; (**b**) broken microcapsule showing the shell structure; (**c**) part of the microcapsule shell with inner surface A and outer surface B; (**d**) elemental analysis of inner surface A; (**e**) elemental analysis of outer surface B.

**Figure 5 materials-15-03708-f005:**
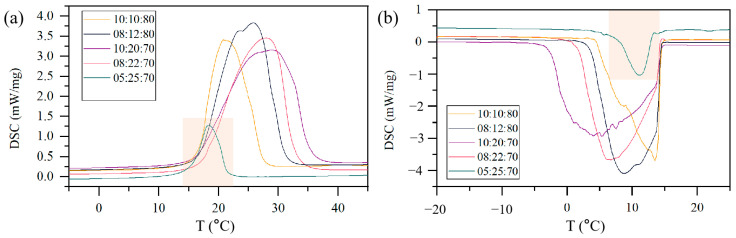
(**a**) Melting and (**b**) crystallization curves of MPCMs with different compositions (IPDI:MMA:C16) of raw materials.

**Figure 6 materials-15-03708-f006:**
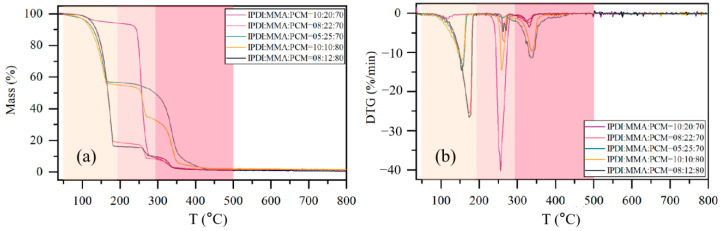
(**a**) TGA thermograms and (**b**) DTG thermograms of MPCMs with different raw material compositions, where the areas of beige, pale pink, and deep pink represent C16 evaporation before 200 °C, evaporation of encapsulated C16 in the range of 200–295 °C, and decomposition of the PU/PMMA shell in the range of 295–500 °C, respectively.

**Figure 7 materials-15-03708-f007:**
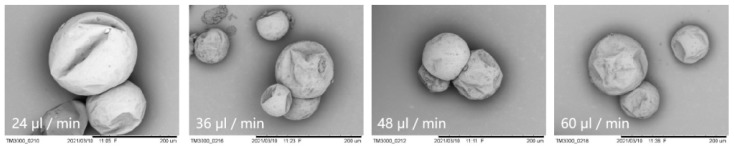
SEM images of MPCMs at different flow rates of the dispersed phase.

**Figure 8 materials-15-03708-f008:**
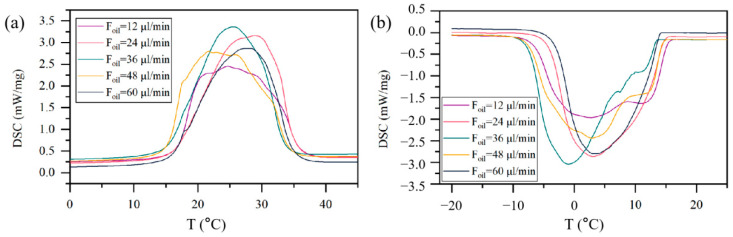
(**a**) Melting and (**b**) crystallization curves of MPCMs prepared at different dispersion phase flow rates.

**Figure 9 materials-15-03708-f009:**
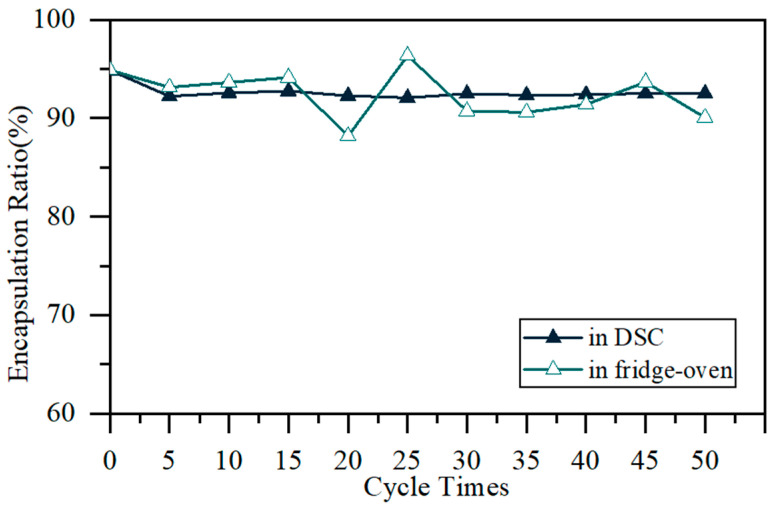
Encapsulation ratio change of an MPCM with IPDI:MMA:C16 = 10:20:70 during 50 DSC cycles and 50 refrigerator-oven cycles.

**Table 1 materials-15-03708-t001:** Thermal properties of MPCMs with different compositions of raw materials.

IPDI:MMA:C16	Melting Enthalpy (J g^−1^)	Crystallization Enthalpy (J g^−1^)	Encapsulation Ratio (%)
n-hexadecane	235.5	231.2	-
10:10:80	147.0	144.5	62.4
8:12:80	209.1	204.5	88.8
10:20:70	209.4	204.7	88.9
8:22:70	210.7	207.3	89.5
5:25:70	34.81	37.94	14.8

**Table 2 materials-15-03708-t002:** Mass fractions of MPCMs calculated from TGA thermograms.

IPDI:MMA:C16	Mass Loss (%)			Residue (%)
	35 °C–200 °C	200 °C–295 °C	295 °C–500 °C	
10:10:80	45	22.34	30.8	1.82
08:12:80	83.91	6.32	9.03	0.73
10:20:70	6.02	84.99	7.99	0.99
8:22:70	81.34	10.57	6.82	1.29
5:25:70	43.47	7	48.4	1.02

**Table 3 materials-15-03708-t003:** Thermal properties of MPCMs prepared at different dispersion phase flow rates.

Sample	Melting Enthalpy (J g^−1^)	Crystallization Enthalpy (J g^−1^)	Encapsulation Ratio C (%)
n-hexadecane	235.5	231.2	-
PUAC16-24	222.6	220.1	94.5
PUAC16-36	218.4	215.9	92.7
PUAC16-48	211.2	208.8	89.7
PUAC16-60	204.7	202.6	86.9

PUAC16-** indicates that C16 is encapsulated into the hybrid shell of PU and PMMA, and the flow rate of the dispersion phase is ** μL min^−1^.

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
