# Peer review of "Microfluidic Fabrication and Thermal Properties of Microencapsulated N-Hexadecane with a Hybrid Polymer Shell for Thermal Energy Storage"

_materials, 2022, doi:10.3390/ma15103708_

Round 1

Reviewer 1 Report

Energy storage is one of the key topic of research today and thermal energy storage requires more attention and caution than electrical energy storage. This paper discuss the thermal energy storage in n-hexadecane PCM which is microencapsulated. This increases the volumetric energy storage capacity of the PCM.

Although similar work are available in literature, they face some issues due to no control on size of droplets. The authors have discussed a fabrication approach where particle size can be tuned. The characterization results verify the stability and energy storage capacity of the fabricated shell.

The data is supported by suitable scientific evidence. The presentation and flow of the manuscript is also appropriate.

Author Response

We greatly appreicate the reviewer for valuable comments.

Reviewer 2 Report

Manuscript ID: materials-1721130

In this article useful information on Microfluidic fabrication and thermal properties of microencapsulated n-hexadecane with a hybrid polymer shell for thermal energy storage has been provided. However, author need to address following comments in order to publish in this journal.

  1. Introduction is very general without any data. Author should add overall energy consumption and emission data. They can refer following most related articles for related information.
  • Muhammad Wakil Shahzad, Muhammad Burhan, Li Ang and Kim Choon Ng, Energy-water-environment nexus underpinning future desalination sustainability, Desalination 413 (2017) 52-64.
  1. They should provide more detain on results.
  2. Figures quality need to improve.
  3. Tables references are missing.
  4. Overall English need to improve.

Article need minor revision for publication

Author Response

We greatly appreciate the reviewer for valuable comments. Point-by-point responses are in the attached file.

Reviewer 3 Report

The manuscript by these Authors deals with a new method proposed to prepare phase change materials with homogeneous and controllable particle sizes. Thermal and spectroscopic characterization was also carried out. The manuscript needs revisions, all my suggestions are reported in the attached pdf.

Author Response

We greatly appreciate the reviewer for valuable comments. Point-by-point responses are in the attached file. 

For other comments listed in the PDF, we appreciate them a lot and made changes in the manuscript according to the comments.

Reviewer 4 Report

  1. The axes of the graphs d and e in Figure 4 are not clear. At the same time, the shell thicknesses specified in the text cannot be understood from the SEM photographs. Figure captions should be legible.
  2. The y-axis of the Figure 9 graph can be scaled more appropriately.
  3. The MMA monomer was used without purification. Is there any inhibitor and retarder in the purchased MMA monomer? Generally, since there are the inhibitors and retarders in monomers, so it is purified by vacuum distillation. The MMA property must be fully specified in the Materials and Methods section.
  4. Figure 1 is the schematic illustration. Instead of this schematic representation, it will be more realistic to put the real picture of the experimental setup.
  5. Articles on the related subject in the years 2020-2022 should be added to the introduction.
  6. The Results and Discussion Section can be developed by comparing the features of MCPMs with literature studies.
  7. English of the manuscript can be improved. Sentences can be shorter and more understandable.

Author Response

(The authors gave the same response as above.)
